# Echocardiographic Evaluation of the Mitral Valve in Cavalier King Charles Spaniels

**DOI:** 10.3390/ani10091454

**Published:** 2020-08-19

**Authors:** Mara Bagardi, Arianna Bionda, Chiara Locatelli, Matteo Cortellari, Stefano Frattini, Alessio Negro, Paola Crepaldi, Paola Giuseppina Brambilla

**Affiliations:** 1Department of Veterinary Medicine, University of Milan, Via dell’Università n. 6, 26900 Lodi, Italy; mara.bagardi@unimi.it (M.B.); arianna.bionda@studenti.unimi.it (A.B.); matteo.cortellari@unimi.it (M.C.); stefano.frattini@unimi.it (S.F.); alessio.negro@unimi.it (A.N.); paola.crepaldi@unimi.it (P.C.); paola.brambilla@unimi.it (P.G.B.); 2Department of Agricultural and Environmental Sciences, University of Milan, Via Giovanni Celoria n. 2, 20133 Milano, Italy

**Keywords:** myxomatous mitral valve disease, Cavalier King Charles Spaniels, echocardiography, anterior mitral valve leaflet

## Abstract

**Simple Summary:**

The incidence of myxomatous mitral valve disease (MMVD) is particularly high in some breeds such as the Cavalier King Charles Spaniel (CKCS) with as many as 90% developing this pathology by the age of 10 years. Our study shows that the American College of Veterinary Internal Medicine (ACVIM) B1 class included extremely different animals. These subjects had significant variability in terms of age (from 6 months to over 11 years), presence and severity of regurgitation and prolapse of mitral valve, as well as presence and intensity of heart murmur. It would therefore seem that in this breed the pathology can essentially follow two different paths: some dogs remain in ACVIM class B1 for a long time (they have a slow progression and persist in class B1 until old age); in others the disease establishes early and progresses rapidly, until it yields symptoms of heart failure even before 8 years. It would therefore be useful to understand whether there are parameters that can early distinguish these two groups.

**Abstract:**

This prospective cross-sectional study aimed to: (1) characterize echocardiographic features of mitral valve in MMVD affected Cavalier King Charles Spaniels (CKCS), focusing on dogs classified as American College of Veterinary Internal Medicine (ACVIM) class B1; (2) compare echocardiographic data in ACVIM B1 dogs divided on the basis of age at time of MMVD diagnosis, in order to understand if different aged subjects had different echocardiographic patterns. Length (AMVL), width (AMVW) and area (AMVA) of the anterior mitral valve leaflet, mitral valve prolapse, diameters of the mitral valve annulus in diastole (MVAd) and systole (MVAs) of 90 CKCS in different ACVIM classes, 64 of which in class B1, were measured. Valvular measurements were indexed to body weight using Wesselowski’s scaling exponents. The presence of heart murmur did not discriminate between A and B1 classes (*p* = 0.128). Heart enlargement was more frequent in males (r^2^ = 0.07, *p* = 0.013). Within class B1, older subjects showed significantly higher values of AMVA, AMVW, MVAd, MVAs and lower sphericity index (SI). Since many CKCS with MMVD have no murmur and their mitral valve has peculiarities, a specifically designed echocardiographic screening should be realized. In addition, different aged B1 dogs have different echocardiographic patterns that may imply different genetic and prognostic profiles.

## 1. Introduction

Myxomatous mitral valve disease (MMVD) is the most commonly acquired heart disease in dogs, accounting for approximately 75% of all dogs with cardiac disease [1], particularly in older dogs and small breeds [2]. Cavalier King Charles Spaniels (CKCS) show the earliest onset and the highest incidence of MMVD, compared with other breeds [1,3,4,5]: it involves almost all subjects over eleven years [6] as well as many dogs under four years of age [7]. At present, there is no treatment for the disease itself that can be applied on a large population of dogs (i.e., valve replacement and repair surgery), so prevention is the only strategy to reduce its incidence.

The etiology of the MMVD has not been fully clarified yet, but it is a common opinion that, at least in some breeds such as CKCS and German Dachshund, the hereditary component plays a predominant role. Other factors, such as exercise level, degree of obesity and diet could contribute to the development of the disease, although with a minor influence [8]. Parental MMVD status is an important factor influencing the probability of the occurrence of heart murmurs and their intensity in offspring [9]. Myxomatous mitral valve disease development is a polygenic threshold trait and the sex of the offspring influences threshold levels: more males than females develop murmurs [9]. This would explain why, in the same family, males generally present the disease at a younger age. Moreover, early MMVD, typically found in CKCS, also appears to be highly heritable [10]. In order to reduce the incidence of the disease, it would be advisable to distinguish CKCS in non-genetically predisposed, predisposed to age-related MMVD (typical of many breeds), and with breed predisposition to its early onset [10]. Up to now, there is no genetic test able to discriminate among these three groups. For this reason, CKCS breeding plans are based only on the cardiac phenotype of the breeders. However, the selection protocols put in place so far in several countries proved to be ineffective when based only on auscultatory findings, while the results were more promising when echocardiographic data were considered as well [11,12,13].

What emerged from all these studies is that, since the prevalence of MMVD is highly dependent on age, it would be important to choose an age limit whereby dogs with early onset MMVD can be excluded from reproduction. On the other hand, to place it at too advanced an age would result in the exclusion of an excessive number of breeders: it is not advisable to exclude more than 30% of dogs from the breeders for a screening program for a single disease [14]. Furthermore, an estimation of the genetic value would allow us to exclude predisposed dogs before their reproduction and to get a predictive evaluation on animals whose health status has not been recorded yet [15].

As reported by Menciotti et al. (2018) [16], a 3D transthoracic echocardiography of the mitral valve of healthy young adult CKCS showed that leaflet tenting and the posterior leaflet were reduced in comparison with dogs of other breeds. These morphologic differences could predispose this breed to the early onset of MMVD [16]. The identification, at a young age, of dogs at high risk of adverse outcomes in the future is desirable [17]. Hence, this breed, in particular subjects in the American College of Veterinary Internal Medicine (ACVIM) class B1, which represent the majority of CKCS undergoing our voluntary recruitment screening program, should be the object of a dedicated study aimed at associating valve morphology with genomics, in order to determine the peculiarities of CKCS who will develop the MMVD earlier and will present an adverse event related to heart disease. Early predictors would also allow for improved breeding recommendations. Previously published data on the echocardiographic anatomy of the mitral valve in dogs are limited to normal valves in seven healthy Norfolk terriers [18] and in sixty MMDV-affected dogs of different breeds [19].

The main objective of this study was to analyze the echocardiographic features of MMVD-affected CKCS in ACVIM B1. We also provided a description of echocardiographic findings related to dogs in different ACVIM classes. Two-dimensional (2-D) echocardiography was chosen for this study, due to its widespread diffusion and easy use by most echocardiographers. Particularly, the mitral valve morphology and the degree of leaflet prolapse were described in order to characterize the echocardiographic anatomy of the CKCS mitral valve apparatus. Secondarily, we compared echocardiographic data in ACVIM class B1 dogs divided according to their age at the time of MMVD diagnosis, in order to understand if different aged subjects had different echocardiographic patterns [20,21]. Our goal is to assess whether there were different echocardiographic patterns within ACVIM class B1. This could lead us to classify valvular lesions and cardiac morphologies typical of the young CKCS predisposed to the early onset of MMVD compared with the older subjects. A part of the results will be given also by future studies of the follow-up of these subjects and by their genomic and morphometric examinations.

## 2. Materials and Methods

This prospective cross-sectional study was carried out including 90 privately owned CKCS visited at the Cardiology Unit of the Veterinary Hospital—Department of Veterinary Medicine, University of Milan, between December 2018 and September 2019. The informed consent was signed by the owners, according to the ethical committee statement of the University of Milan number 2/2016. Each dog underwent a clinical and cardiological examination (including a complete echocardiographic exam). Dates of birth were verified by checking each animal’s microchip number in the regional registry. Auscultatory findings were evaluated by three well-trained operators: the presence/absence, timing, intensity (0 = absent; 1 = I-II left systolic; 2 = III-IV bilateral systolic; 3 = V-VI bilateral systolic), and the point of maximum intensity of the murmur were recorded. Blood pressure was indirectly measured with a Doppler method according to the ACVIM consensus statement [22].

Echocardiographic examination (2-D, M-mode, spectral, and color-flow Doppler) was performed using MyLab50 Gold cardiovascular ultrasound machine (Esaote, Genova, Italy) equipped with multi-frequency phased array probes (3.5–5 and 7.5–10 MHz), chosen according to the weight of the subject. The exam was performed conforming to a standard procedure [23]. Video clips optimized for the visualization of mitral valve apparatus were acquired and stored using the echo machine software for offline measurements. All measurements of interest were repeated by one single operator (MB) on three consecutive cardiac cycles, and the mean value was used in the statistical analysis [24].

Measurements of the same stored images were repeated in a randomly selected subset of six CKCS by MB one week after the initial measurements. These data were used to assess intra- and inter-observer repeatability. Diagnosis of MMVD was based on 2-D and color Doppler echocardiographic findings: typical lesions of the mitral valve apparatus and a demonstrated mitral regurgitation (MR) on the color Doppler echocardiogram were considered as the definitive diagnostic criteria [24]. The right parasternal four-chamber long-axis view is considered the standard view for the assessment of the canine mitral valve [19,25,26,27] and of the mitral annulus [19].

The following measures were obtained from this view: the sphericity index (SI), the length (AMVL), width (AMVW) and area (AMVA) of the anterior mitral valve leaflet, and diameters of the mitral valve annulus in diastole (MVAd) and systole (MVAs). Moreover, the degree of mitral valve (MV) prolapse and regurgitation were studied. The SI was calculated as the ratio of the left ventricle (LV) long-axis diameter to short-axis diameter in end-diastole [28] and used as indicator of LV remodeling. The AMVL, AMVW and AMVA were measured (in centimeters) during diastole, when the leaflet was fully extended, whereas the MVAd and MVAs in the first frame, respectively, after closing (end-diastole) and before the opening (end-systole) of the leaflets [19]. Particularly, MVAd and MVAs were obtained by measuring the distance from mitral valve hinge point to hinge point, with frame-by-frame advancement utilized to accurately identify their location [19]. All these measurements (AMVL, AMVW, AMVA, MVAd and MVAs) were indexed to body weight using the scaling exponents calculated by Wesselowski for each specific valve measure [19]. They were, respectively: 0.37, 0.41, 0.78, 0.37 and 0.40.

Mitral valve prolapse was considered mild if the leaflets were prolapsing but did not cross the line joining their pivotal points (line P), moderate if they protruded between the P line and the line joining half of the echoic areas located in the lower part of the atrial septum at the level of atrioventricular junction (T line), and severe if the leaflets exceeded the T line [29] (Figure 1). Lastly, MR was assessed by color Doppler, calculating the maximal ratio of the regurgitant jet area signal (ARJ) to the left atrium area (LAA) (ARJ/LAA ratio) in left parasternal long-axis view [24].

Regurgitation was considered mild if it occupied less than one third of the atrium, moderate if between one and two thirds, severe if more than two thirds [24]. The following measurements were taken from the right parasternal short-axis view: the left atrial to aortic root ratio (LA/Ao) was obtained by 2-D technique, whereas the left ventricular diameters were measured in M-mode according to the leading-edge-to-leading-edge method [30,31].

Left ventricular end diastolic (EDV) and end systolic volumes (ESV) were calculated by the Teichholtz method and values were successively indexed for body surface area (BSA) to obtain the end-diastolic (EDVI) and the end-systolic (ESVI) volume indexes [32]. The left ventricular internal diameters in diastole and systole normalized for body weight (LVIDdN and LVIDsN, respectively) were calculated using the allometric equation, as previously described [33]. Left ventricular fractional shortening (%FS) was calculated using the formula:[(LVIDd-LVIDs)/LVIDd] × 100(1)

Trans-mitral flow [E peak velocity (E-Vmax), A peak velocity (A-Vmax), E-Vmax to A-Vmax ratio (E/A)] was measured using pulsed-wave Doppler (PWD) from the left four-chamber apical view.

Dogs were staged and stratified according to ACVIM guidelines [34]. Subjects in class B1 were also allocated in age-related groups, according to the age at time of MMVD diagnosis: up to three years (group 1), between three and six years (group 2) and over six years old (group 3).

### 2.1. Analysis of Genealogical Data

The pedigrees were obtained consulting the online genealogy book of the Ente Nazionale della Cinofilia Italiana (ENCI) (https://www.enci.it/libro-genealogico/libro-genealogico-on-line, accessed 18 August 2020) or the Livre des Origines Français (LOF) select of the Centrale Canine site (https://www.centrale-canine.fr/lofselect, accessed 18 August 2020). The genealogical data were reported on an Excel worksheet and then analyzed with OPTISEL^®^, a library of the free downloadable R^®^ program (https://CRAN.R-project.org/package=optiSel, accessed 18 August 2020). The following parameters were analyzed: inbreeding coefficient (F) (probability that both alleles inherited from an individual are copies of a single allele from an ancestor common to both parents), average relatedness coefficient (AR) (probability that a randomly chosen allele in the whole pedigree population belongs to a given animal), pedigree completeness (proportion of known ancestors in each generation) and depth of the pedigree defined by the number of fully traced generations (in which all the ancestors are known) and the number of maximum generations traced (in which at least one ancestor is known).

### 2.2. Statistical Analysis

Statistical analyses were performed using SPSSTM 26.0 (IBM, Armonk, New York, USA). Descriptive statistics were generated. The distribution of data for continuous variables was assessed for normality by means of the Kolmogorov–Smirnov test. The variables were reported as the mean ± standard deviation in cases of normal distribution, and otherwise as the median and interquartile range (IQR—from the 25th to 75th percentile). The correlation was considered weak, moderate, or strong, respectively, when the value of the correlation coefficient was less than 0.3, between 0.3 and 0.7, or more than 0.7. The associations between continuous and categorical variables were investigated with the χ2 test and their strength was evaluated with the coefficient of determination (r^2^). The comparison between continuous and categorical variables was assessed with the Kruskal–Wallis test and the significance values were adjusted according to the Bonferroni correction. The intra-observer and inter-observer repeatability of all variables obtained from the four-chamber right parasternal long-axis image plane (AMVL, AMVW, AMVA, MVAd and MVAs) were further characterized through the calculation of coefficients of variation (CV%).

## 3. Results

### 3.1. Clinical and Genealogical Results

Ninety CKCS were included; 28 dogs belonged to private owners and 62 to eight different breeders. The population consisted of 60 females (26 neutered), and 30 males (two neutered). Age ranged from 0.5 to 11.7 years (5.67 ± 2.75 years). Males’ and females’ mean ages were not significantly different. The weight ranged from 5.1 to 12.8 kg (9.13 ± 1.94). Weight was mildly positively related to age (r^2^ = 0.18, *p* = 0.000). Mean weight in males (9.72 ± 0.35 kg) was significantly higher (*p* = 0.042) than in females (8.84 ± 0.25 kg). Among the 90 dogs included in the study, 62 (68.9%) were registered into the ENCI website or Centrale-canine database. For 28 subjects (31.1%) it was not possible to find any information about their genealogy. The pedigree analysis of the 62 pedigrees identified a total of 1207 animals, consisted of the dogs in our sample and all their known ancestors. The number of fully traced generations of the 62 pedigrees of our study’s dogs was, on average, 3.77, ranging from two to five and the number of maximum generations traced was on average 10.61, ranging from six to 14. Average relatedness coefficient mean values of all 1207 subjects and of the 62 dogs included in this study were, respectively, 0.006 and 0.055. The mean F value for all 1207 subjects was 0.005, ranging from zero and 0.25, whereas for the 62 dogs included in this study the mean F was 0.018. There was not a significant association between F and all analyzed variables.

Eighty-one (90%) dogs were affected by MMVD. The subjects were classified as follows: nine (10%) in ACVIM class A, 64 (71%) in class B1, 11 (12%) in class B2, six in class C/D (7%). In 59% of included dogs a heart murmur was detected at auscultation; heart murmur intensity was highly positively correlated with ACVIM class (r^2^ = 0.75, *p* = 0.000) and moderately with the age (r^2^ = 0.61, *p* = 0.000). There were no discrepancies in the three operators’ assessments of the presence/absence and intensity of heart murmurs. The presence of a murmur did not discriminate between A and B1 classes (*p* = 0.128): in 44% of B1 subjects, a heart murmur was not audible. In 76% of dogs without a heart murmur, there was an echocardiographic diagnosis of MMVD. Age was moderately positively related to ACVIM class (r^2^ = 0.52, *p* = 0.000). Although sex was not related to the presence of MMVD (*p* = 0.456) and ACVIM class (*p* = 0.083), heart chamber enlargement (class B2 and C/D) was statistically more frequent in males (r^2^ = 0.93, *p* = 0.02). The severity of mitral regurgitation was highly significantly related to the ACVIM class (r^2^ = 0.78, *p* = 0.000).

### 3.2. Echocardiographic Data

Regarding the echocardiographic measurements, the intra-observer and the inter-observer (MB vs. CL) coefficients of variation (CV range in %) were <10% and <20%, respectively, for each tested variable. Table 1 shows the LA/Ao, E/A, EDVI, ESVI, FS%, LVIDdN and LVIDsN of all included subjects expressed in relation to ACVIM class. In affected dogs, LA/Ao and LVIDdN were significantly greater (respectively, *p* = 0.011 and *p* = 0.017) in males than in females; in healthy dogs, no differences were observed between sexes.

In the whole population, 83 (92.2%) had a MV prolapse: 42 dogs showed a mild, 35 a moderate and six a severe prolapse. The anterior leaflet was always involved; in 77.1% of cases, the prolapse also affected the posterior one. No significant correlations were found between prolapse severity and ACVIM class (*p* = 0.210) or MR severity (*p* = 0.310).

Table 2 shows all indexed mitral valve measurements and their significant differences in relation to each ACVIM class. Anterior mitral valve leaflet length (AMVL) was longer in class B2 compared to A and in classes B2 and C/D compared to class B1. The same was observed for AMVW and AMVA: they were greater in classes B2 and C/D than in classes A and B1. Mitral valve annuli in diastole (MVAd) and systole (MVAs) were larger in class C/D than in classes A and B1. No statistically significant differences were found for mitral valve measurements between A and B1 classes and between B2 and C/D classes. Moreover, MVAd (*p* = 0.004) and MVAs (*p* = 0.003) were larger in males and MVAd was significantly larger in neutered females than in all females (*p* = 0.007).

Clinical and echocardiographic parameters of subjects in ACVIM class B1 divided according to age-related classes are reported in Table 3. The results of the statistical analysis of these variables are reported in Table 4.

Subjects in class B1 were divided into age-related classes (age at time of MMVD diagnosis): up to 3 years (group 1), between 3 and 6 years (group 2) and over 6 years old (group 3). Ten dogs were allocated (15.6%) to group 1, 30 (46.9%) to group 2 and 24 (37.5%) to group 3. Anterior mitral valve width (AMVW) and area (AMVA) were greater in group 3 than group 1. Mitral valve annulus in diastole (MVAd) was greater in group 3 compared to group 1 and 2, whereas MVAs was greater in group 3 compared to group 1 only. No significant differences in mitral valve measurements were found between different sexes in ACVIM class B1.

## 4. Discussion

The prevalence of MMVD in our population was 90%. Despite this high prevalence, the subjects presenting cardiac remodeling and related symptoms were, respectively, 14% and 7%. Due to the small sample size dimension of subjects in ACVIM classes A, B2 and C/D, only the results obtained for ACVIM class B1 can be considered as statistically significant, whereas other data should be considered only descriptive of the screened population. Of the included CKCS, 80% were referred for echocardiographic screening and this may explain the high number of B1 subjects presented at the visit by breeders. All symptomatic subjects were at least 7.5 years old. Despite the small number of subjects in ACVIM class B2 and above, as expected, the presence and severity (ACVIM class) of the disease were positively related to age [7,14]. Patients aged over 7 years were all affected, but the disease was also diagnosed in very young dogs (50% of those under 2 years of age, all classified as ACVIM B1). These results are consistent with those reported by other authors, which indicated valvular alterations in 67% of dogs aged from 6 months to 3 years and 95% of older dogs [7]. Weight was positively related to ACVIM class (*p* = 0.013). This result contrasts with two different studies, which observed a negative correlation [15]. However, it should be considered that in these studies the significance of the correlation was mild and in our case the weight explained only 9% of the variability of the ACVIM classification. This association was probably affected by the positive correlation between the weight and age of the subjects (r^2^ = 0.42, *p* = 0.000). It is very interesting to note that 64% of the included dogs weighed over 8 kg, the upper limit accepted by the ENCI standard for the CKCS [35]; however, it was not possible to distinguish overweight dogs due to a lack of evaluation of subjects’ body condition score (BCS). The observation of a weight greater than the ENCI standard may also be due to the recruitment of companion subjects and not only of those bred for reproduction and exposition; however, it would be advisable to investigate this issue by increasing the sample size, to determine whether the values obtained are associated with an overweight problem or whether the morphology of many CKCS differs from the breed standard.

In this study, contrary to those reported in the literature [1], the disease did not have a higher prevalence in males and affected males were not significantly younger than affected females. This could be justified by the higher number of females presented for breeder screening. Nevertheless, despite the small number of subjects in ACVIM classes B2 and C/D, the risk of remodeling in affected males was higher than in females: there were more males than females in ACVIM class B2 or above and affected males showed higher values of LA/Ao and LVIDdN. As reported by Misbach et al. (2014) [31], there were no significant differences between sexes in healthy dogs. These data are also to be considered only as descriptive and not as statistically significant, given the scarcity of healthy subjects included in this study.

Genealogy information could not be obtained for approximately 1/3 of the dogs included in the study. Although the number of unregistered subjects is not negligible, it should be noted that none of them was under five years of age: this could indicate that, in Italy, the purchase of dogs without a pedigree has decreased in the last few years. The average inbreeding coefficient (F) of our sample population and their ancestors, equal to 0.5%, was lower than the one reported in the literature, which ranged from a minimum of 0.9% in Belgium to a maximum of 6.3% in CKCS born in 2018 in the UK [36]. The low relatedness among our subjects should ensure that our findings are not related to single families and popular sires’ peculiarities.

In 59% of the dogs, it was possible to hear a heart murmur during auscultation. Despite the association between murmur intensity and ACVIM class, the auscultation was not an eligible method for discriminating between A and B1 subjects. These data underline the importance of performing echocardiographic screening, especially in this breed. In support of this, it should be noted that the selection protocols put in place in several countries proved to be ineffective when based only on auscultatory findings, while, when taking into account the echocardiographic data, the results were more promising [11,12,13].

The description of mitral valve anterior leaflets in each ACVIM class and particularly in the B1 class allowed us to delineate quantitative echocardiographic characteristics of a population of CKCS affected by this pathology. Increases in the thickness, length and area of the anterior mitral valve leaflet were evident in more severely affected patients (ACVIM classes B2 and C/D). These echocardiographic changes are parallel to the classic valvular remodeling reported on gross pathologic examination [37]. Additionally, there was an increase in the diameter of the MVAd in patients with more advanced MMVD, as reported by Wesselowski et al. (2015) [19]. All these findings are consistent with the expected changes associated with chronic mitral regurgitation, volume overload and enlargement of the left heart, despite the small size of B2 and C/D subjects [37]. The AMVL and AMVW obtained in this study were greater than those reported by Wesselowski et al. (2015) for each ACVIM class [19]: this is in accordance with the valvular characteristics peculiar to CKCS [16]. This finding is even more relevant considering that annular measurements were consistent with Wesselwoski’s, underlining a mitral valve dimension proportionally greater than the heart one. Furthermore, the results of this study underline how the valvular measurements varied considerably within the same class (ACVIM B1), which showed clinical and echocardiographic characteristics of extreme heterogeneity. Firstly, heart murmur severity was associated with the age of the subjects and with the severity of mitral valve morphology alterations. Secondly, annular dilation was not expected in ACVIM B1 subjects, as this group was defined by the absence of chamber enlargement, but a modification of the annulus in older dogs was observed, although remaining within the standard limits. Further investigations are needed to understand if this would predict a worse evolution of the disease. Similarly, in older B1 CKCS, a significant reduction of the SI was observed (more spherical left ventricle), although it was not associated with a significant increase in the left atrial and ventricular dimensions. This, as described by Sargent et al. in 2015 for dogs affected by MMVD at different ACVIM stages, would be predictive of cardiac mortality. Further investigations with a larger study population could help to clarify whether additional echocardiographic valvular changes can be identified in CKCS in ACVIM class B1 and an appropriate follow-up would allow us to highlight prognostic factors related to disease worsening within this ACVIM class.

To summarize, in this study, 79% of the dogs were classified as B1 and this class included extremely different animals. In addition to the small heterogeneity in terms of MV morphology, ACVIM B1 subjects had significant variability in terms of age (from six months to over eleven years), the presence and severity of mitral valve regurgitation and prolapse and presence and intensity of heart murmur. Therefore, it would seem that, in this breed, the pathology can essentially follow two different paths. In the first case, dogs are predisposed to age-related MMVD (typical of many breeds) and have a slow progression of the disease. In the second one, the disease onsets early and progresses rapidly, so that symptoms of heart failure could develop even before 8 years of age. It would therefore be useful to understand whether there are parameters that can early distinguish these two groups. In our opinion, these parameters could be associated with morphological, echocardiographic, and genetic data. Therefore, the number of young symptomatic dogs and mildly affected old ones will be increased and the follow-up of ACVIM classes A and B1 will be performed in order to observe the pathology progression and to also relate it to a morphologic evaluation. The improvement of ACVIM class A subjects will be useful to identify reference intervals in healthy CKCS for the echocardiographic performed measures. Thanks to the characterization of mitral valve morphology and prolapse, we were able to identify case and control subjects for a preliminary genetic analysis that will investigate MMVD-related genes and pathways that are the base of the predisposition and early onset of MMVD in this breed. In this way, we will be able to address, based on clinical, echocardiographic, and genetic evidence, breeding programs that also include MMVD.

This study is not without its limitations. The main limitation was the uneven numerosity of the ACVIM classes. In this regard, increasing the number of subjects in the different classes would allow a more appropriate delineation of the clinical and echocardiographic profile of each of them. It must be highlighted that the literature does not report reference intervals for healthy CKCS. Secondly, there was a disproportion in the number of females compared to males: this is due to the voluntary adherence to the screening program and to the greater participation of breeders compared to private owners. Furthermore, the lack of assessment of the subjects’ BCS, as well as the lack of morphometric data, did not allow a proper evaluation of their weight and therefore a more precise indexing of the echocardiographic measurements [15,38]. Lastly, the use of only the right parasternal four-chamber view for the identification of the mitral regurgitation area in this study is considered as a limitation.

## 5. Conclusions

In conclusion, this is the first study that describes measurements of the anterior mitral valve leaflet and the mitral valve annulus in the CKCS affected by MMVD at different stages. It must be highlighted that, due to the small sample size of ACVIM class A, B2 and C/D, only data regarding B1 subjects can be considered as statistically significant. Regarding ACVIM class B1 dogs, the diameter of the mitral valve annulus in systole and diastole, as well as the thickness and area of the anterior mitral valve leaflet and the sphericity of the left ventricle, are greater in patients with MMVD diagnosed at an advanced age. The follow-up of these dogs will be essential to uniquely associate ventricular and valvular echocardiographic features with MMVD progression. A future aim will also be the evaluation of the posterior mitral valve leaflet, both in healthy and affected CKCS.

## Figures and Tables

**Figure 1 animals-10-01454-f001:**
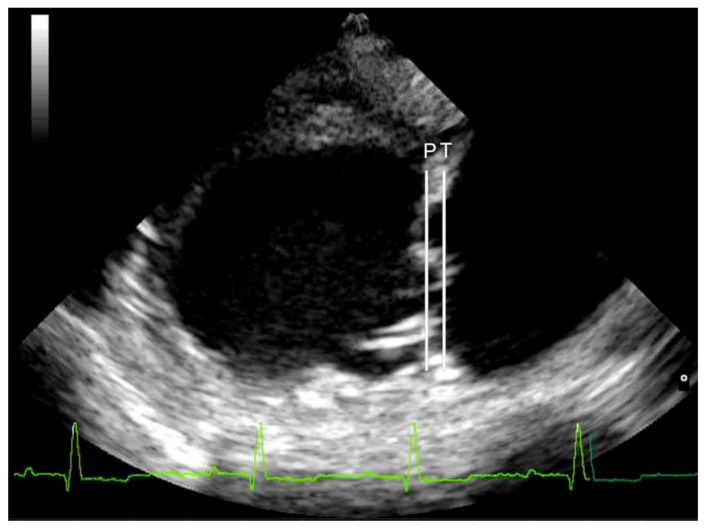
Right parasternal long-axis four-chamber view of a Cavalier King Charles Spaniel (CKCS) affected by myxomatous mitral valve disease (MMVD) in American College of Veterinary Internal Medicine (ACVIM) class D; severe mitral valve prolapse was assessed by the protrusion of one or both leaflets over line T. The arrows outline the severity of mitral valve prolapse and the affected leaflet. Line P is drawn from the hinge point of the anterior leaflet to the hinge point of the posterior leaflet. Line T is drawn from the middle of the elliptical echogenic area at the lower part of the atrial septum to the atrioventricular junction, i.e., the junction between the left ventricular wall/annulus fibrosis and the left atrial wall.

**Table 1 animals-10-01454-t001:** Clinical and echocardiographic parameters of all included subjects related to ACVIM classes. Females (F); males (M); n. of subjects with heart murmur in each ACVIM class (HM); left atrium to aorta ratio (LA/Ao); E and A waves ratio (E/A); end-systolic volume index (ESVI); end-diastolic volume index (EDVI); shortening fraction (%FS); left ventricular internal diameter in diastole normalized for body weight (LVIDdN); left ventricular internal diameter in systole normalized for body weight (LVIDsN). The variables are reported as the mean and standard deviation (SD) in cases of normal distribution, and otherwise as the median and interquartile ranges (IQR—from 25th to 75th).

STAGE (n.)		F	M	Age (y)	Weight (kg)	HM	LA/Ao	E/A	ESVI (mL/m^2^)	EDVI (mL/m^2^)	%FS	LVIDdN (cm/kg)	LVIDsN (cm/kg)
A (n. 9)	n.	7	2	9	9	0	9	9	9	9	9	9	9
	Mean			2.86	7.41						35.11		
	SD			1.89	1.13						9.23		
	Median						1.19	1.24	14.28	41.22		1.22	0.76
	IQR						1.11–1.33	1.09–1.35	9.95–18.2	37.88–53		1.17–1.33	0.67–0.84
B1 (n. 64)	n.	46	18	64	64	36	64	62	64	64	64	64	64
	Mean			5.36	9.23						34.42		
	SD			2.56	1.77						6.15		
	Median						1.23	1.30	20.99	55.28		1.36	0.89
	IQR						1.11–1.31	1.16–1.42	16.1–25.93	48.2–69.77		1.29–1.49	0.8–0.95
B2 (n. 11)	n.	4	7	11	11	11	11	11	11	11	11	11	11
	Mean			8.31	9.51						42.82		
	SD			1.61	2.29						6.78		
	Median						1.70	1.44	27.88	118.46		1.85	0.98
	IQR						1.59–1.65	1.31–1.46	22.72–34.93	104.65–113.64		1.82–1.94	0.90–1.07
C/D (n. 6)	n.	3	3	6	6	6	6	6	6	6	6	6	6
	Mean			8.49	10.02						46.16		
	SD			0.78	2.82						4.62		
	Median						1.96	1.59	39.28	182.74		2.21	1.13
	IQR						1.81–2.28	1.62–2.18	31.72–49.20	149.01–241.52		2.04–2.50	1.03–1.25

**Table 2 animals-10-01454-t002:** Indexed mitral valve measurements of all analyzed subjects related to ACVIM classes. All measurements were indexed to body weight using the scaling exponents calculated by Wesselowski, one for each specific valve measure (Wesselowski et al. 2015). They were, respectively: 0.37, 0.41, 0.78, 0.37 and 0.40 for anterior mitral valve length (AMVL), width (AMVW), area (AMVA), mitral valve annulus in diastole (MVAd) and systole (MVAs). Sphericity index (SI). The variables were not normally distributed and are reported as the median and interquartile ranges (IQR—from 25th to 75th). For n. 1 subject in ACVIM class A was not possible to measure the anterior mitral valve leaflet. * Within a column, values differ significantly (*p* < 0.01) from ACVIM class B1. † Within a column, values differ significantly (*p* < 0.01) from ACVIM class A.

			AMVL (cm/kg)	AMVW (cm/kg)	AMVA (cm/kg)	MVAd (cm/kg)	MVAs (cm/kg)	SI
STAGE	A	n.	8	8	8	8	8	8
		Median	0.75 †	0.14	0.09	0.76	0.56	1.38
		IQR	0.66–0.79	0.11–0.15	0.07–0.10	0.68–0.87	0.52–0.64	1.31–1.50
	B1	n.	62	61	61	64	64	64
		Median	0.69	0.15	0.09	0.85	0.62	1.38
		IQR	0.63–0.77	0.12–0.17	0.07–0.11	0.75–0.93	0.55–0.73	1.24–1.48
	B2	n.	11	11	11	11	11	11
		Median	0.91 *	0.19^*^†	0.13 *†	0.94	0.71	1.27
		IQR	0.87–0.96	0.18–0.21	0.11–0.15	0.85–1.03	0.64–0.79	1.17–1.36
	C/D	n.	6	6	6	6	6	6
		Median	0.94 *	0.22 *†	0.16 *†	1.23 *†	1.00 *†	1.15 *†
		IQR	0.87–1.00	0.19–0.31	0.13–0.21	1.13–1.44	0.71–1.24	1.03–1.25

**Table 3 animals-10-01454-t003:** Statistically significant differences between clinical findings, indexed echocardiographic measurements and SI of subjects in ACVIM class B1 categorized considering the age at time of MMVD diagnosis. Age groups: group (1) under 3 years (n.10–15.6%); group (2) between 3 and 6 years (n. 30–46.9%); group (3) over 6 years old (n.24–37.5%). The significance values have been adjusted according to the Bonferroni correction for more tests. Sex: entire male (eM), neutered male (nM); entire female (eF); neutered female (nF). Heart murmur severity: 0=absent; 1=I-II left systolic; 2=III-IV bilateral systolic; 3=V-VI bilateral systolic; left atrium to aorta ratio (LA/Ao); E and A waves ratio (E/A); end-systolic volume index (ESVI); end-diastolic volume index (EDVI); shortening fraction (%FS); left ventricular internal diameter in diastole normalized for body weight (LVIDdN); left ventricular internal diameter in systole normalized for body weight (LVIDsN); anterior mitral valve length (AMVL), width (AMVW), area (AMVA), mitral valve annulus in diastole (MVAd) and systole (MVAs); sphericity index (SI); mitral regurgitation severity(MR) (0=absent; 1=regurgitant jet area (ARJ)/left atrial area (LAA) < 1/3; 2 = 1/3 < ARJ/LAA < 2/3; 3 = ARJ/LAA > 2/3); mitral valve prolapse severity (MVP) (0 = absent, 1 = under P line, 2 = between P and T lines, 3 = over T line). The variables are reported as the mean and standard deviation (SD), and as the number of subjects for each grade of severity for heart murmur, MR and MVP. * Within a column, values differ significantly (*p* < 0.01) from group 1. † Within a column, values differ significantly (*p* < 0.01) from group 2.

Sex	Age (y)	Heart Murmur Severity	Weight (kg)	ESVI (mL/m^2^)	EDVI (mL/m^2^)	LA/Ao	E/A	%FS	LVIDdN (cm/kg)	LVIDsN (cm/kg)	MR	MVP	SI	AMVL (cm/kg)	AMVW (cm/kg)	AMVA (cm/kg)	MVAd (cm/kg)	MVAs (cm/kg)
**Group 1**																		
4eM, 6eF	1.90 ± 0.77	0 (n.9)1 (n.1)	7.85 ± 1.25	20.73 ± 5.97	56.30 ± 18.40	1.13 ± 0.13	1.44 ± 0.24	32.20 ± 5.01	1.36 ± 0.17	0.88 ± 0.09	0 (n.3) 1 (n.7)	0 (n.1) 1 (n.7) 2 (n.2)	1.36 ± 0.27	0.70 ± 0.07	0.13 ± 0.02	0.08 ± 0.02	0.79 ± 0.12	0.62 ± 0.09
**Group 2**																		
15eF, 7nF, 7eM, 1nM	4.45 ± 1.03	0 (n.13) 1 (n.16) 2 (n.1)	9.36 ± 1.55 *	20.32 ± 7.78	57.77 ± 16.24	1.21 ± 0.13	1.33 ± 0.24	34.23 ± 6.76	1.37 ± 0.16	0.86 ± 0.13	0 (n.4) 1 (n.25) 2 (n.1)	1 (n.22) 2 (n.7) 3 (n.1)	1.40 ± 0.21	0.71 ± 0.13	0.15 ± 0.04	0.09 ± 0.03	0.82 ± 0.11	0.60 ± 0.11
**Group 3**																		
6eF, 12nF, 5eM, 1nM	8.07 ± 1.63	0 (n.6) 1 (n.11) 2 (n.7)	9.97 ± 2.18 *	22.80 ± 8.25	64.90 ± 19.96	1.26 ± 0.13	1.21 ± 0.21	35.58 ± 5.70	1.43 ± 0.17	0.90 ± 0.13	1 (n.17) 2 (n.6) 3 (n.1)	0 (n.1) 1 (n.9) 2 (n.14)	1.22 ± 0.3 †	0.72 ± 0.11	0.16 ± 0.03 *	0.10 ± 0.03 *	0.90 ± 0.09 *†	0.69 ± 0.10 †

**Table 4 animals-10-01454-t004:** Statistically significant differences between clinical findings, indexed echocardiographic measurements and SI of subjects in ACVIM class B1, categorized considering mitral regurgitation, mitral valve prolapse and heart murmur severity. Mitral regurgitation severity: 0=absent; 1=regurgitant jet area (ARJ)/left atrial area (LAA) < 1/3; 2 = 1/3 < ARJ/LAA < 2/3; 3 = ARJ/LAA > 2/3. Mitral valve prolapse severity: 0=absent; 1=under P line; 2=between P and T lines; 3=over T line. Heart murmur severity: 0 = absent; 1=I-II left systolic; 2=III-IV bilateral systolic; 3=V-VI bilateral systolic. The significance values have been adjusted according to the Bonferroni correction for more tests. left ventricular internal diameter in diastole normalized for body weight (LVIDdN); anterior mitral valve length (AMVL), width (AMVW), area (AMVA), mitral valve annulus in diastole (MVAd) and systole (MVAs); sphericity index (SI).

**Pairwise Comparison Mitral Regurgitation Severity in ACVIM Class B1**			
	Severity Class	Sign.	Adapted Sign.
Age (y)	0–2	0.001	0.008
**Pairwise Comparison Mitral Valve Prolapse Severity in ACVIM Class B1**			
	Severity Class	Sign.	Adapted Sign.
Age (y)	1–2	0.002	0.009
**Pairwise Comparison Heart Murmur Severity in ACVIM Class B1**			
	Severity Class	Sign.	Adapted Sign.
Age (y)	0–1	0.005	0.016
	0–2	0.000	0.001
E/A	0–1	0.004	0.012
LVIDdN (cm/kg)	0–2	0.000	0.001
AMVW (cm/kg)	0–2	0.005	0.014
	1–2	0.007	0.021
AMVA (cm/kg)	0–2	0.001	0.003
	1–2	0.001	0.004
MVAd (cm/kg)	0–2	0.010	0.030
MVAs (cm/kg)	0–1	0.008	0.024
	0–2	0.002	0.006
SI	2–0	0.011	0.032

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
