# Peer review of "Echocardiographic Evaluation of the Mitral Valve in Cavalier King Charles Spaniels"

_animals, 2020, doi:10.3390/ani10091454_

Round 1

Reviewer 1 Report

Myxomatous mitral valve disease (MMVD) is the most common acquired heart disease in dogs, occurring with a high frequency particularly in small breeds. The onset of disease in Cavalier King Charles Spaniels (CKCS) is early and, data suggest a hereditary involvement in its development.

The findings of this paper consist to characterize echocardiographic features of the mitral valve in Cavalier King Charles Spaniel, comparing the differences between healthy and myxomatous mitral valve disease-affected dogs. The Authors have investigated an interesting topic, but the sample size of dogs - included in ACVIM class A, B2, C and D - is very low to consider the publication in this form.

Although the objectives of the paper are of interest and fit well within the scope of the journal, the introduction provides sufficient background, the methods have been properly described and the quality of language is sufficient, as well as the figure and the tables, the Authors should increase the case study.

In my opinion, the manuscript could be accepted for publication in Animals with the above-mentioned suggestions.

Author Response

Reviewer 1

Comments and Suggestions for Authors

Animals-890114. Echocardiographic evaluation of the mitral valve in 2 Cavalier King Charles Spaniels

The authors thank the Editor and the Reviewers for their thoroughly review of our study.

We have carefully considered all Reviewers’ comments and have tried to address them whenever we felt this was appropriate. We feel that the quality of our manuscript has improved following the Reviewers’ comments and suggestions.

Best regards

Myxomatous mitral valve disease (MMVD) is the most common acquired heart disease in dogs, occurring with a high frequency particularly in small breeds. The onset of disease in Cavalier King Charles Spaniels (CKCS) is early and, data suggest a hereditary involvement in its development.

The findings of this paper consist to characterize echocardiographic features of the mitral valve in Cavalier King Charles Spaniel, comparing the differences between healthy and myxomatous mitral valve disease-affected dogs. The Authors have investigated an interesting topic, but the sample size of dogs - included in ACVIM class A, B2, C and D – is very low to consider the publication in this form.

Thank you very much for your revision and for this suggestion. We are conscious that the disproportion of sample size in each ACVIM class could have been a problem. For this reason, given the short time available to implement the sample size, we decided to focus this work only on subjects in ACVIM class B1 and still leave the echocardiographic description of the subjects in other ACVIM classes (considering C and D subjects as singular group as suggested by Reviewer 2). All analyses have been performed again considering the new classification (C/D). Furthermore, we declare in the discussion paragraph that due to the small sample size dimension of subjects in ACVIM classes A, B2 and C/D, only the results obtained for ACVIM class B1 must be considered as statistically significant. We have also modified the discussion in order to improve this concept.

Although the objectives of the paper are of interest and fit well within the scope of the journal, the introduction provides sufficient background, the methods have been properly described and the quality of language is sufficient, as well as the figure and the tables, the Authors should increase the case study.

In my opinion, the manuscript could be accepted for publication in Animals with the above-mentioned suggestions.

Thank you so much for appreciating our work. This makes us very grateful.

As suggested, given the disproportion of the subjects, we decided to focus only on the class ACVIM B1 and we declare it both in abstract and text, in particular in discussion paragraph.

Reviewer 2 Report

Animals-890114. Echocardiographic evaluation of the mitral valve in 2 Cavalier King Charles Spaniels

General comments:  This prospective cross-sectional study echocardiographically examined the width, length, and area of the anterior mitral valve leaflet, mitral valve prolapse, diameters of the mitral valve annulus in diastole/systole in 90 CKCS. The Authors are commended for addressing an important subject; however, this reviewer has some concerns/suggestions for the Authors, as listed below.

Specific comments:

Simple Summary

Line 8: “…….heart failure even before 8.” 8 weeks, months, years…???? Specify, please.

Abstract:

Lines 22-24: The first aim is: “characterize echocardiographic features of mitral valve in healthy and MMVD affected CKCS, focusing on dogs classified as ACVIM class B1”. This reviewer disagrees with the Authors: they have included only 9 CKCS in class A ACVIM…….to characterize echocardiographic features of mitral valve in healthy CKCS, Authors need to enroll more dogs. Modify the first aim, please.

Line 28: “….90 CKCS in different ACVIM classes….”. Authors should underline the number of CKCS in class B1…..this is the main aim of the study.

Introduction

Line 82: See comments above.

Moreover, Authors should emphasize the use of 2-D transthoracic echocardiography to obtain the echocardiographic features in their study respect to previous study (Menciotti et al, 2018).

Materials and Methods

Line 101-103: “The cardiovascular system was evaluated by three well-trained operators: the presence/absence, timing, intensity (grade I-VI/VI), and the point of maximum intensity of the murmur were recorded.”. These are auscultatory findings, cardiovascular system evaluation allows to record more clinical findings. Specify, please.

Line 108: “High-quality video clips…”. What does it mean?

Moreover, in this reviewer’s experience, optimized MV images can be obtain with the image sector narrowed to include only the MV apparatus or using a zoom to maximize image resolution.

Line 110: “..one single operator..”. Who?

Line 113: “These data were used to assess intra-observer repeatability”. Concerning inter-observer repeatability? This should be assessed.

Line 115-116: “…mitral valve apparatus and a demonstrated mitral regurgitation (MR) on the color Doppler echocardiogram…”. Did the Authors use the right parasternal four-chamber long axis view to assess the MV regurgitation? Previous studies have indicated that detection of MV regurgitation is more sensitive from the left apical view……..This could be a limitation of the study.

Lines 123-126: It is unclear, when the Authors perform the AMVL, AMVW and AMVA measurements.

Results

Line 203: “…5 in class C (6%) and 1 in class D (1%).” Only 1 dog in class D………consider creating a only group of 6 dogs in class C/D.

Line 205: “…..heart murmur intensity was moderately positively correlated with ACVIM class (r2=0.44, p=0.000) and with the age (r2=0.29, 205 p=0.000).”. Authors consider moderately correlated values of r2=0.44 and r2=0.29……….Can they provide a reference for this and report in the statistical analysis paragraph, please?

Lines 206-207: “There were no discrepancies in the three operators’ assessment of presence/absence and intensity of heart murmurs.” It is unclear, how did the Authors assess this? And the results?

Line 216: “..(CV range in %)….”. This should be stated in the Materials and Methods - statistical analysis paragraph.

Table 3: “Heart murmur severity: 0=absent; 1=I-II left systolic; 2=III-IV bilateral systolic; 3=V-VI bilateral systolic.”…………..This scale of severity appears in this table for the first time! State this in the Materials and Methods paragraph, please.

Discussion/limitations

Authors should report in the limitations the absence of reference interval in healthy CKCS for the echocardiographic features assessed in this study. Moreover, could be useful assess also posterior leaflet of the MV in the future studies.

Author Response

Reviewer 2

Comments and Suggestions for Authors

Animals-890114. Echocardiographic evaluation of the mitral valve in 2 Cavalier King Charles Spaniels

The authors thank the Editor and the Reviewers for their thoroughly review of our study.

We have carefully considered all Reviewers’ comments and have tried to address them whenever we felt this was appropriate. We feel that the quality of our manuscript has improved following the Reviewers’ comments and suggestions.

Best regards

General comments: This prospective cross-sectional study echocardiographically examined the width, length, and area of the anterior mitral valve leaflet, mitral valve prolapse, diameters of the mitral valve annulus in diastole/systole in 90 CKCS. The Authors are commended for addressing an important subject; however, this reviewer has some concerns/suggestions for the Authors, as listed below.

Thank you so much for appreciating our work. This makes us very grateful. Our answers to your comments are given below.

Specific comments:

Simple Summary

Line 8: “…….heart failure even before 8.” 8 weeks, months, years…???? Specify, please.

Thank you, we added “years” in order to be more specific. Thank you.

Abstract:

Lines 22-24: The first aim is: “characterize echocardiographic features of mitral valve in healthy and MMVD affected CKCS, focusing on dogs classified as ACVIM class B1”. This reviewer disagrees with the Authors: they have included only 9 CKCS in class A ACVIM…….to characterize echocardiographic features of mitral valve in healthy CKCS, Authors need to enroll more dogs. Modify the first aim, please.

Thank you very much for this suggestion, which agrees with that of the other reviewer. We are conscious that the disproportion of sample size in each ACVIM class could have been a problem. For this reason, given the short time available to implement the sample size, we decided to focus this work only on subjects in ACVIM class B1 and still leave the echocardiographic description of the subjects in other ACVIM classes (considering C and D subjects as singular group). All analyses have been performed again considering the new classification (C/D). Furthermore, we declare in the discussion paragraph that due to the small sample size dimension of subjects in ACVIM classes A, B2 and C/D, only the results obtained for ACVIM class B1 must be considered as statistically significant.

Line 28: “….90 CKCS in different ACVIM classes….”. Authors should underline the number of CKCS in class B1…..this is the main aim of the study.

Thank you very much. Thank to this comment the abstract is more focalized on ACVIM B1 subject, giving a specific introduction to our work.

Introduction

Line 82: See comments above. Moreover, Authors should emphasize the use of 2-D transthoracic echocardiography to obtain the echocardiographic features in their study respect to previous study (Menciotti et al, 2018).

Thank you for this suggestion. We added a speculation about the widespread use of 2-D echocardiography respect to 3-D.

Materials and Methods

Line 101-103: “The cardiovascular system was evaluated by three well-trained operators: the presence/absence, timing, intensity (grade I-VI/VI), and the point of maximum intensity of the murmur were recorded.”. These are auscultatory findings, cardiovascular system evaluation allows to record more clinical findings. Specify, please.

Thank you, this is a mistake. For this reason, we have rephrased the sentence (lines 102-104) and added the severity classification (lines 103-104).

Line 108: “High-quality video clips…”. What does it mean? Moreover, in this reviewer’s experience, optimized MV images can be obtain with the image sector narrowed to include only the MV apparatus or using a zoom to maximize image resolution.

Thank you very much for this comment. We were referring to video clips optimized for the visualization of the mitral valve apparatus, both in zoom mode and not. We have replaced this terminology (lines 109-111).

Line 110: “..one single operator..”. Who?

Thank you, we have added this information. This was a lack (line 111).  

Line 113: “These data were used to assess intra-observer repeatability”. Concerning inter-observer repeatability? This should be assessed.

Thank you very much for this suggestion. This was not our focus and for this reason we did not test inter-observer variability. We decide to perform inter observer repeatability during this week on a subset of 6 subjects randomly chosen and we have added this information and related results.

Line 115-116: “…mitral valve apparatus and a demonstrated mitral regurgitation (MR) on the color Doppler echocardiogram…”. Did the Authors use the right parasternal four-chamber long axis view to assess the MV regurgitation? Previous studies have indicated that detection of MV regurgitation is more sensitive from the left apical view……..This could be a limitation of the study.

Thank you very much for this comment. We have always performed a double check of MR in both right and left view. Otherwise, we decided to use the method described by Chetboul (2012) for our classification. We have added this limitation in the discussion (lines 389-390). Thank you very much.

Lines 123-126: It is unclear, when the Authors perform the AMVL, AMVW and AMVA measurements.

Thank you very much. It was a lack to not specify the timing of this measurement. We based our work and our measurements according to Wesselowski (2015) method. For this reason, we performed AMVL, AMVW and AMVA during diastole, when the leaflet was fully extended. We have added this information and we have rephrased this sentence (lines 125-127).

Results

Line 203: “…5 in class C (6%) and 1 in class D (1%).” Only 1 dog in class D………consider creating a only group of 6 dogs in class C/D.

Thank you for this suggestion, thank to this comment we have decides to describe these subjects a singular group (line 208).

Line 205: “….. heart murmur intensity was moderately positively correlated with ACVIM class (r2=0.44, p=0.000) and with the age (r2=0.29, 205 p=0.000).”. Authors consider moderately correlated values of r2=0.44 and r2=0.29………. Can they provide a reference for this and report in the statistical analysis paragraph, please?

Thank you, this is a mistake. We have just specified in statistical analysis the reference values for this classification (lines 181-182) and we have corrected the r2=0.29 as “slightly” (line 210).

Lines 206-207: “There were no discrepancies in the three operators’ assessment of presence/absence and intensity of heart murmurs.” It is unclear, how did the Authors assess this? And the results?

Thank you very much for this comment. We have not specified this point because when a Cavalier King Charles Spaniel entered our clinic, it has been always listened by MB, CL and PGB for a blind evaluation of the concordance of auscultation. A specific statistic has not been performed as the evaluations always agreed (lines 211-212).

Line 216: “..(CV range in %)….”. This should be stated in the Materials and Methods - statistical analysis paragraph.

Thank you for this comment, this was a lack. As stated before, we have performed the evaluation of inter-observer variability between MB and CL. We have also added this information in material and methods paragraph (lines 186-189).

Table 3: “Heart murmur severity: 0=absent; 1=I-II left systolic; 2=III-IV bilateral systolic; 3=V-VI bilateral systolic.”…………..This scale of severity appears in this table for the first time! State this in the Materials and Methods paragraph, please.

Thank you, we added this information in material and methods paragraph (lines 103-104).

Discussion/limitations

Authors should report in the limitations the absence of reference interval in healthy CKCS for the echocardiographic features assessed in this study. Moreover, could be useful assess also posterior leaflet of the MV in the future studies.

Thank you very much for this suggestion.

Due to the small sample size dimension of subjects in ACVIM classes A, B2 and C/D, we declare in the discussion paragraph that only the results obtained for ACVIM class B1 must be considered as statistically significant and that other data must be considered only descriptive for screened population (lines 287-289).

As stated before, we have added in the limits paragraph the only use of right parasternal four chamber view for the identification of mitral regurgitation area and the absence of reference interval in healthy CKCS for the echocardiographic performed measures.

As future aim we add the evaluation of posterior mitral valve leaflets.
